# Unveiling the Genome of the Diploid Wild Sugarcane Relative *Narenga porphyrocoma* (Hance) Bor

**DOI:** 10.3390/ijms26136124

**Published:** 2025-06-26

**Authors:** Haibi Li, Yiyun Gui, Jinju Wei, Kai Zhu, Hui Zhou, Ronghua Zhang, Dongliang Huang, Sijie Huang, Shuangcai Li, Jisen Zhang, Yangrui Li, Xihui Liu

**Affiliations:** 1Guangxi Key Laboratory of Sugarcane Genetic Improvement, Ministry of Agriculture and Rural Affairs, Sugarcane Research Institute, Guangxi Academy of Agricultural Sciences, Nanning 530007, China; lihaibi@gxaas.net (H.L.); guiyiyun@gxaas.net (Y.G.); jjwei@gxaas.net (J.W.); zhukai@gxaas.net (K.Z.); zhouhui@gxaas.net (H.Z.); zhangronghua@gxaas.net (R.Z.); huangdl@gxaas.net (D.H.); huangsijie@gxaas.net (S.H.); snilsc@126.com (S.L.); 2Guangxi South Subtropical Agricultural Science Research Institute, Guangxi Academy of Agricultural Sciences, Chongzuo 532415, China; 3State Key Laboratory for Conservation and Utilization of Subtropical Agro-Bioresources, College of Agriculture, Guangxi University, Nanning 530004, China; zjisen@126.com

**Keywords:** *Narenga porphyrocoma* (Hance) Bor, genome assembly, drought resistance, gene annotation, sugarcane breeding

## Abstract

*Narenga porphyrocoma* (Hance) Bor is a close relative of sugarcane, with traits such as drought resistance, robustness, early maturity, and disease resistance. In this study, we report the first genome assembly of *N. porphyrocoma* (Hance) Bor GXN1, a diploid species with a chromosomal count of 2*n* = 30. We assembled the genome into 15 pseudochromosomes with an N50 of 128.80 Mp, achieving a high level of completeness (99.0%) using benchmarking universal single-copy orthologs (BUSCO) assessment. The genome was approximately 1.8 Gb. Our analysis identified a substantial proportion of repetitive sequences, primarily long terminal repeats (LTRs), contributing to 69.12% of the genome. In total, 70,680 protein-coding genes were predicted and annotated, focusing on genes related to drought resistance. Transcriptome analysis under drought stress revealed the key gene families involved in plant physiological rhythms and hormone signal transduction, including aquaporins, late embryogenesis abundant proteins, and heat shock proteins. This research reveals the genome of the diploid wild sugarcane relative *N. porphyrocoma* (Hance) Bor, encouraging future studies on gene function, genome evolution, and genetic improvement of sugarcane.

## 1. Introduction

Sugarcane (*Saccharum* spp.) is the world’s largest crop by harvest volume, cultivated over approximately 26 million hectares globally, according to data from the Food and Agriculture Organization of the United Nations [1]. It is an important raw material accounting for 80% of sugar and 40% of bioethanol production, exceeding 1.83 billion tons produced annually (https://www.fao.org/faostat/zh/#data/QV, accessed on 28 April 2025). The genus *Saccharum* comprises six polypoid species, including two wild (*Saccharum spontaneum* and *S. robustum*) and four cultivated species (*S. sinense*, *S. barberi*, *S. officinarum*, and *S. edule*) [2]. These species and their close relatives, *Erianthus* Michx., *Sclerostachya* (Hach.) A. Camus, *Narenga* Bor, and *Miscanthus* Anderss, which contribute key resistance traits, form the “*Saccharum* complex” [3,4].

Sugarcane breeding has a history of >130 years, beginning in Java and Barbados in 1887. Currently, commercially cultivated varieties are complex hybrids of two or more species [5]. The concept of sugarcane “nobilization” was established in 1987 [6]. Crossing the noble canes *S. officinatum* and *S. barberi* with *S. spotaneum* successfully combined the noble traits of high-sugar, high-yield, and stress-resistance genes, resulting in superior varieties such as POJ2878, which are highly resistant to Sereh disease and are widely cultivated globally [7]. This approach resulted in a hybrid genome comprising a mix of aneuploid and homologous chromosomes, ranging from 100 to 130 depending on the specific cross [8,9]. Approximately 70–80% of these chromosomes originate from *S. officinarum*, 10–20% from *S. spontaneum*, and approximately 10% result from interspecific recombination [10,11,12]. However, limited breeding combinations have led to a narrow genetic base, restricting further breeding advancement. Thus, expanding the genetic diversity by incorporating new beneficial genes is crucial for modern sugarcane breeders.

*N. porphyrocoma* (Hance) Bor is a close relative of sugarcane, with traits such as drought resistance, robustness, early maturity, and disease resistance. It has a chromosome number of 2*n* = 30 and is considered the closest diploid relative to sugarcane [13]. In addition, based on an estimation of the molecular clock of the *Adh* gene, it may have diverged from sugarcane approximately 2.5 million years ago [14]. The use of *N. porphyrocoma* (Hance) Bor for breeding was first reported in 1952, which produced hybrids with chromosome counts of 2*n* = 95 and 55 [15]. Studies on *N. porphyrocoma* (Hance) Bor have been conducted for germplasm collection, ploidy identification [16], gene cloning related to tillering, genetic diversity analysis, and molecular identification [17], resulting in F_1_, BC_1_, and BC_2_ hybrids [18,19,20]. *N. porphyrocoma* (Hance) Guangdong64 is diploid (2*n* = 2x = 30), with 10 chromosomes derived from *S. officinarum* and the remaining five newly recombined chromosomes [9]. Owing to the narrow genetic background of modern sugarcane varieties, crossing with wild germplasm such as *Narenga porphyrocoma* (Hance) Bor—which not only possesses excellent agronomic traits but also has fewer chromosomes—offers a valuable breeding strategy, as its simpler genome facilitates more efficient trait introgression, improves meiotic stability in hybrids, and streamlines molecular breeding efforts.

The application of genomics in crops provides substantial opportunities for trait screening, marker-assisted breeding, and parentage testing. It also enables pan-genomic studies in identifying core and indispensable genes in crops. As *N. porphyrocoma* (Hance) Bor shows exceptional drought resistance, identifying its key drought-related genes is beneficial for future agricultural prospects. By analyzing the transcriptome of *N. porphyrocoma* (Hance) Bor subjected to 22 d of drought stress, 104,644 genes were identified, with 21.40% matching sorghum genes. Functional analysis indicated that these genes were enriched in “plant physiological rhythms” and “plant hormone signal transduction”. Several key gene families were identified, including aquaporins, late embryogenesis abundant (LEA) proteins, auxin-related proteins, transferrins, heat shock proteins, and LHCB. These data also revealed novel genes encoding *LHCB*, *DIVARICATA*, and heat stress transcription factors, providing new insights into the regulatory network of *N. porphyrocoma* (Hance) Bor under drought stress [16]. Owing to the complex polyploid nature of sugarcane, a limited number of genomes of hybrid sugarcane, such as SP80-3280 [21], KK317 [22], R570 [23], and a chromosome-level genome of cultivar ZZ1 [24], have been published. The present study presents the first genomic assembly of *N. porphyrocoma* (Hance) Bor. This well-assembled genome reveals the chromosomal location of abscisic acid (ABA) signaling genes, which may guide future drought-resistant breeding programs.

## 2. Results

### 2.1. Genome Assembly and Annotation

To assemble a draft genome of *N. porphyrocoma* (Hance) Bor GXN1, we acquired 1023.75 Gb subreads of PacBio HiFi sequencing and 363.43 Gb Hi-C clean reads of DNB-seq (Appendix A). After clustering, anchoring, and orientation using valid chromatin interactions from Hi-C data, they were anchored into 15 pseudochromosomes with N50 = 128.81 Mb (Figure 1A,B). The chromosomes were arranged and numbered based on the related species of the genus *Saccharum* (chromosome base number x = 10). The accuracy and completeness of the final genome assembly were evaluated. Compared to the core gene sets of the BUSCO database, the completeness proportion was 99.0% (Table 1). In addition, in situ hybridization of *N. porphyrocoma* (Hance) Bor GXN1 chromosomes was performed using 5S rDNA and 45S rDNA probes (Figure 1C–E). Two 45S rDNA sites were present, one of which was located on a satellite chromosome. Additionally, four 5S rDNA sites were distributed across the two different chromosomes. Brighter 5S rDNA sites were found on the longer chromosomes, whereas lighter 5S rDNA sites were found on the shorter chromosomes. The chromosomes exhibited significant variation in length and were non-homologous. Therefore, *N. porphyrocoma* (Hance) Bor GXN1 was classified as diploid (2*n* = 2x = 30). Flow cytometry analysis confirmed that *N. porphyrocoma* (Hance) Bor GXN1 was diploid (Appendix A). Based on k-mer analysis, heterozygosity was approximately 2.27%, with a genome size of approximately 1.80 Gb (Appendix A).

### 2.2. Repetitive Sequence Identification and Annotation

Repetitive sequences are identical or symmetrical DNA fragments within the genome that not only contribute to genome structure but also play critical roles in gene regulatory networks, influencing gene expression, chromatin organization, and genome stability. Owing to their indispensable roles in gene expression and transcriptional regulation and their influence on evolution, heredity, and variation, repetitive sequences are primarily categorized into tandem and interspersed repeats. Tandem repeats include microsatellites and minisatellites, while interspersed repeats, also known as transposable elements, include Type I retrotransposons (transcribed into RNA and then reverse-transcribed into DNA) and Type II DNA transposons (which do not undergo reverse transcription). Common retrotransposons include LTRs, long interspersed nuclear elements (LINEs), and short interspersed nuclear elements (SINEs).

Homology-based alignment and de novo prediction were used to annotate repetitive sequences. For the homology-based alignment approach, we employed RepeatMasker [25] and RepeatProteinMask to identify and classify sequences similar to known repetitive sequences in the RepBase database. For the de novo prediction approach, we used RepeatModeler v2.0.1 [26] and LTRharvest [27] to build de novo repetitive sequence libraries; then, using RepeatMasker, we annotated the repetitive sequences [25]. Finally, the Tandem Repeats Finder [28] was used to detect tandem repeats in the genome. Compared to homology-based alignment, de novo prediction can predict transposable elements based on their structural features without depending on existing transposable element databases, thus allowing the discovery of novel transposable elements. In total, 1259 Mb (69.12%) of the genome assembly were identified as repetitive sequences, and retrotransposons with a total length of 911.979 Mb accounted for 50.02% of the whole genome (Table 2). The two most frequent LTR types were Copia and Gypsy, accounting for 13.19% and 30.93% of the genome, respectively (Table 2). Transposable elements accounted for only 9.25% of the genome (Table 2). Furthermore, 15 putative centromeric regions were detected on the celery chromosomes (Figure 2).

### 2.3. Gene Prediction and Annotation

The RNA-Seq data (SRA633912) were downloaded from NCBI, each RNA-Seq dataset was aligned to the genome sequence using HISAT2 [29], and transcript assembly was performed using StringTie v3.0.0 [30]. TransDecoder (v5.5.0) with default settings was used to predict ORFs from the obtained RNA-seq transcripts. Using annotation information from eight related species (*Z. mays* B73, *B. stacei*, *E. rufipilus*, *O. sativa*, *S. spontaneum* Np-X, *Setaria italica*, and *Sorghum. bicolor*), homology-based gene structure predictions were conducted using GeMoMa [31]. Additionally, de novo prediction was performed using Augustus [32] based on the species training set. To obtain the final gene sets, the annotation results obtained from the transcriptome, homology, and de novo methods were integrated using the EVidenceModeler v1.1.1 [33]. Finally, we evaluated the quality of the gene set by comparing it with that of the BUSCO database. In total, 70,860 genes were annotated using these methods, and the average gene length was 3362.29 bp, with an average of 4.67 exons per gene (Table 3).

Using the gene set obtained from gene structure annotation, we employed DIAMOND [34] to obtain functional information from known protein databases, including NR, Swiss-Prot, KEGG, KOG, TrEMBL, InterPro, and GO. Overall, 70,033 (99.08%) *N. porphyrocoma* (Hance) Bor GXN1 genes were annotated, and approximately 36,681 genes were annotated by all five databases (Figure 3A and Appendix A). KEGG enrichment analysis was conducted to analyze the gene families in *N. porphyrocoma* (Hance) Bor GXN1 (Figure 3B). Carbohydrate metabolism, environmental adaptation, and signal transduction were the most enriched pathways, indicating the potential of *N. porphyrocoma* (Hance) Bor GXN1 to be a valuable genetic resource for sugar production and ecological adaptability.

### 2.4. Constructing Ancestral Karyotypes and Deducing Chromosome Change Trajectories

Chromosome fission and fusion are essential processes in genome evolution that frequently result in significant structural changes and genetic diversity. Notably, five of the *N. porphyrocoma* (Hance) Bor GXN1 chromosomes showed syntenic signals with the other ten chromosomes, suggesting that these five chromosomes originated from the fission and fusion of ten chromosomes (Figure 4). Chromosome 11 shared orthology with chromosomes 1 and 8, indicating that it may have originated from the fusion or fission of these two chromosomes. Similarly, chromosome 12 comprised fragments of chromosomes 2 and 6, suggesting a fusion event. Chromosome 13 represented a combination of chromosomes 3 and 5, chromosome 14 was formed from chromosomes 4 and 7, and chromosome 15 was formed from the fusion of chromosomes 10 and 9 (Figure 4). These fission and fusion events reflect the dynamic nature of chromosomal evolution, contributing to genetic variability and potentially influencing phenotypic traits. Understanding these processes is essential to gain insights into the evolutionary history and genetic complexity of species. The fusion of two different chromosomes is a common chromosomal rearrangement found in species within the *Saccharum* complex. This is consistent with karyotype analysis based on the chromosome-specific painting of the *Saccharum* complex [9].

### 2.5. Evolutionary History of N. porphyrocoma (Hance) Bor GXN1

To understand the genomic changes that occurred pre- and post-lineage divergence, we calculated the synonymous nucleotide substitutions (Ks) values among homoeologous gene pairs from *N. porphyrocoma* (2*n* = 2x = 30), *Sorghum bicolor* (2*n* = 2x = 20), *M. sinensis* (2*n* = 2x = 38), *E. rufipilus* (2*n* = 2x = 20), and *S. spontaneum* Np-X (2*n* = 4x = 40), and we then plotted the Ks distributions for all syntelogues. In this study, the inter- and intra-species divergence times were consistent with those of previous studies [35]. The divergence time analysis of *N. porphyrocoma*, *E. rufipilus*, and *S. spontaneum* Np-X indicated that their divergence occurred within a short timeframe (Table 4 and Figure 5). Compared to the interspecies Ks peaks for *E. rufipilus* and *S. spontaneum* Np-X, *N. porphyrocoma* (Ks = 0.002) had a small Ks peak, suggesting that its polyploidization likely occurred after divergence from *E. rufipilus* and *S. spontaneum* Np-X at approximately 0.15 Mya.

### 2.6. Gene Family Analysis Regarding Drought Stresses

Drought stress, characterized by water deficit in crops, leads to reduced or total loss of crop yields. Genetic variability drives adaptive responses to drought and other stresses [36]. Conventional breeding has traditionally been the primary approach for generating genetic variability in sugarcane for varietal improvement [37]. However, progress has been hindered by the genetic complexity of sugarcane [38]. We consider the importance of the abscisic acid (ABA) signaling pathway, particularly the roles of the mitogen-activated protein kinase (MAPK), protein phosphatase 2C (PP2C), pyrabactin resistance-like (PYL), and LEA gene families in drought resistance through their involvement in a complex regulatory network. We analyzed these gene families in *N. porphyrocoma* (Hance) Bor GXN1 and its relatives (Table 5). Our analysis revealed that, except for CC-01-1940, these gene families showed no substantial gene expansion or contraction according to their ploidy (Table 5). Furthermore, we mapped the MAPK, PP2C, PYL, and LEA families across 15 chromosomes (Figure 6). This mapping provided valuable information to design molecular markers for the tracking of these genes in future breeding programs.

## 3. Discussion

Sugarcane is a crucial crop for the production of sugar and bioethanol worldwide. Owing to the complicated polyploid nature of the sugarcane genome, limited well-assembled genomes have been identified. In addition to the breeding program designed to achieve “noble traits” in sugarcane, modern sugarcane varieties have very narrow genetic bases. In the present study, we obtained the first genome assembly of *N. porphyrocoma* (Hance) Bor GXN1 using PacBio HIFI sequencing and Hi-C to generate high-quality reference assembly and annotation. PacBio HIFI sequencing enabled a highly continuous de novo genome assembly. The high BUSCO completeness score (99%) demonstrates the advantages of Hi-C scaffolding for whole-genome assembly by capturing proximal interactions to correct and merge contigs, thereby minimizing fragmentation and achieving high-quality assembly. Several studies have incorporated three-dimensional chromatin conformation analyses, such as Hi-C and Pore-C, to improve the scale of traditional phased linear de novo assemblies [39]. Combining PacBio HIFI and Hi-C sequencing with less sequencing data provides an economical method to efficiently assemble the genome.

Based on chromosomal paintings, *N. porphyrocoma* (Hance) Bor GXN1 var. Guangdong64 (2*n* = 2x = 30) has a genomic composition of ten chromosomes derived from *S. officinarum*, with the remaining five chromosomes being novel recombinations [9]. In this study, we used comprehensive genomic data to reconstruct the ancestral karyotypes of *N. porphyrocoma* (Hance) Bor GXN1 and elucidate the trajectories of its chromosomal evolution. We identified syntenic signals indicating that five chromosomes originated from the fission and fusion of 10 ancestral chromosomes, consistent with the dynamic chromosomal rearrangements observed in the Saccharum complex (Figure 4). Our analyses provided comprehensive insights into the origins and structural rearrangements of chromosomes in this species. Comparative genomic investigations identified the contributions of *S. officinarum* and clarified the mechanisms underlying the formation of recombined chromosomes. These findings are consistent with the chromosomal painting results, validating the hybrid origin of *N. porphyrocoma* var. Guangdong64. This integrative approach enhances the understanding of chromosomal evolution and provides a basis for future research on genomic diversity and crop improvement.

Based on the molecular clock analysis of the *Adh* gene, previous studies have proposed that *N. porphyrocoma* (Hance) Bor GXN1 diverged from sugarcane approximately 2.5 million years ago [14]. However, our investigation using calculated Ks values among homoeologous gene pairs—inter- and intra-species—revealed that polyploidization in *N. porphyrocoma* (Hance) Bor GXN1 likely occurred more recently, post-diversion from *E. rufipilus* and *S. spontaneum* Np-X. As more sugarcane-related species will be sequenced in the future, we anticipate a more refined reconstruction of the evolutionary history and chromosomal arrangements within this lineage. This will provide comprehensive insights into the mechanisms driving the speciation and genomic evolution of sugarcane and related species.

ABA is a critical hormone that acts as the primary signal during drought stress in plants, where it is synthesized in the roots and transported to the shoots. Therein, it plays a major role in regulating stomatal closure to minimize water loss via transpiration [40,41]. At the molecular level, ABA signaling involves the activation of the SNF1-related protein kinase 2 family, particularly subclass III, through the PYR/PYL/RCAR-PP2C receptor complex. This leads to the phosphorylation of several downstream transcription factors, such as AREB1, AREB2, ABF1, and ABF3, which are crucial for ABA-dependent gene regulation during drought [42]. Several MAPKs are directly or indirectly activated by the ABA core signaling pathway [19]. Furthermore, ABA regulatory genes, such as ERA1, have been manipulated to enhance drought tolerance in crops such as sugarcane [43]. We analyzed the core ABA signaling pathway gene families using the well-assembled *N. porphyrocoma* (Hance) Bor GXN1 genome. Our analysis revealed that *N. porphyrocoma* possesses a comparable number of ABA-related genes (e.g., MAPK, PP2C, PYL, and LEA families) to its relatives, without significant expansion or contraction relative to ploidy. This finding aligns with the work of Singh and Laxmi (2015) [42], who emphasized the importance of ABA-regulated transcriptional networks in drought responses. In addition, by mapping these genes to their chromosomal locations, we provide a valuable resource for marker-assisted breeding aimed at enhancing drought tolerance in sugarcane.

The integration of *N. porphyrocoma* into sugarcane breeding programs offers a promising strategy to expand genetic diversity and introduce traits such as drought resistance, robustness, and disease resistance. Its diploid genome simplifies trait introgression and improves meiotic stability in hybrids, addressing the challenges posed by the polyploid nature of modern sugarcane. This approach is supported by recent successes in utilizing wild relatives for crop improvement, as demonstrated by Bao et al. (2024) [24] in their chromosomal-scale genome assembly of hybrid sugarcane.

## 4. Materials and Methods

### 4.1. Sample Collection, DNA Extraction, and Sequencing

*N. porphyrocoma* (Hance) Bor GXN1 plants were collected from a barren mountain near Nanning, China (22°53′06.7″ N, 108°21′36.6″ E), and transplanted into a greenhouse at the Guangxi Academy of Agricultural Sciences, Nanning, China. Genomic DNA was extracted from young leaves of three individual plants, and libraries were prepared following the PacBio standard single-molecule real-time (SMRT) bell construction protocol. The libraries were sequenced on the PacBio Sequel II platform, and the resulting HiFi reads were processed using SMRTLink (parameters: min passes = 3, min rq = 0.99) to remove adapter sequences and low-quality or short reads.

Additionally, a Hi-C library was prepared using the Proximo Hi-C plant methodology (https://info.phasegenomics.com/protocols, accessed on 10 June 2023) developed by Phase Genomics (Seattle, WA, USA) and sequenced on the DNBSEQ T7 platform. Adapters and duplicate reads were removed, and the remaining reads were filtered to retain only those with a sequencing quality score > Q30, yielding the final raw reads.

### 4.2. Genome Assembly and Assessment

PacBio HiFi reads were initially assembled using Hifiasm v0.15, with the parameters “–write-ec–write-paf-u-l0” [44]. Redundant sequences were further trimmed using Purge Haplotigs [45], and Hi-C data were used to anchor the contigs to chromosomes.

To identify genuine interaction pairs, Hi-C reads were filtered using FASTP v0.20.0 [46], and valid mapped read pairs were selected using the Juicer pipeline [47]. A chromosome-level assembly was constructed using 3D-DNA v201013 (https://github.com/aidenlab/3d-dna, accessed on 15 October 2023).

The genome assembly quality was evaluated using benchmarking universal single-copy orthologs (BUSCO) v4.05 [48] with the “embryophyta_odb10” ortholog set. To ensure high fidelity and completeness, HiFi reads were mapped back to the genome using Minimap2 [49], verifying the alignment accuracy of the sequencing data with the assembly. The final assembly accuracy was further validated using Merqury with 21-mers.

### 4.3. Repeat Sequences and Gene Annotations

A de novo repeat library was constructed using RepeatModeler v2.0.1 [26] and primarily long terminal repeat (LTR) harvest with default settings. Repetitive sequences were predicted using RepeatMasker v4.0.7 [25] and a de novo repeat library. Non-redundant repetitive sequences were compiled by merging the results of the two methods. Tandem Repeat Finder v4.09 [28] was used to identify tandem repeats.

Gene prediction in the repeat-masked genome was performed using homology search, reference-guided transcriptome assembly, and ab initio prediction. For homology-based prediction, GeMoMa v1.9 [31] was used to align homologous peptides from *Brachypodium stacei*, *Erianthus rufipilus*, *Oryza sativa*, *S. spontaneum* Np-X, *Setaria italica*, *S. bicolor*, and *Zea mays* B73 for assembly, and gene structure information was extracted for homolog prediction. RNA-seq-based prediction was conducted by aligning the filtered mRNA-seq data (national center of biotechnology information [NCBI]: SRR6322472 and SRR6322473) with the reference genome using HISAT2. StringTie v1.3.4d [30] was used to assemble transcripts, and TransDecoder (https://github.com/TransDecoder/TransDecoder, accessed on 22 June 2025) was used to identify the open reading frames (ORFs). For de novo prediction, a training dataset was generated and used for ab initio gene prediction using Augustus v3.4.0 [50] under default settings. Gene predictions from all the methods were integrated using the EVidenceModeler v1.1.1 [51], which filters out miscoded genes. TransposonPSI (http://transposonpsi.sourceforge.net/, accessed on 22 June 2025) was used to eliminate transposable element genes.

Functional annotations of the predicted protein-coding genes were validated against multiple databases. InterProScan v5.36 [51] was used to identify signal peptides, transmembrane domains, protein families, motifs, domains, and gene ontology (GO) terms. BLASTP v2.7.1 [52] was run with an E-value threshold of 1 × 10^−5^ against Swiss-Prot, non-redundant database (NR), translation of EMBL (TrEMBL), KOG [53], Kyoto encyclopedia of genes and genomes (KEGG) [54], and GO [55] databases. The top hits from these searches were combined to finalize the gene annotations and provide comprehensive functional information for the predicted protein-coding genes.

### 4.4. Annotation of Noncoding RNAs

Noncoding RNA sequences in the genome were annotated using database searches and model-based predictions. INFERNAL v1.1.5 [56] was employed to query the Rfam database and identify the sequences of miRNAs, rRNAs, small nuclear RNAs, and small nucleolar RNAs. tRNAscan-SE v2.0.12 [57] was used to predict tRNA sequences based on eukaryotic parameters, whereas RNAmmer v1.2 [58] was used to detect rRNAs and their subcomponents.

### 4.5. Divergence Time Analysis

Gene sets from *N. porphyrocoma*, *Sorghum bicolor*, *Miscanthus sinensis*, *E. rufipilus,* and *S. spontaneum* were obtained from publicly available databases and preprocessed to ensure sequence quality and completeness. All sequences were subjected to quality trimming, contaminant removal, and redundancy filtering to ensure that only high-quality sequences were used in subsequent analyses. Whole-genome duplication (WGD) software [59] was used to perform comparative genomics analysis and align and cluster the gene sets among the five species. This process facilitated the identification of paralogous gene pairs within *N. porphyrocoma* and orthologous gene pairs between species, which are crucial for elucidating the evolutionary relationships and impact of WGD on the gene repertoire. After the identification of orthologous and paralogous gene pairs, we calculated the synonymous substitution rate (Ks) for each pair. Ks measures the number of synonymous substitutions per site and indicates the time since the genes diverged from a common ancestor. Ks was calculated using the built-in functionalities of WGD software, which applies established algorithms for the estimation of synonymous substitutions. To infer the time of divergence between species and the occurrence of WGD events in *N. porphyrocoma*, the following formula was applied.T = Ks/2λ × 10^−6^ Mya(1)
where T represents the time in million years ago (Mya), Ks is the synonymous substitution rate per site, and λ is the average substitution rate. We used the average substitution rate of 6.5 × 10^−9^, as reported [53].

### 4.6. Fluorescent In Situ Hybridization (FISH) Analysis

The root tips were fixed in Carnoy’s solution and subjected to a series of ethanol treatments. A cell suspension was prepared by enzymatic digestion and applied to slides for FISH using 45S and 5S rDNA probes labeled with Cy3 and FAM, respectively. The slides were subjected to a pretreatment regimen, including pepsin digestion and ethanol dehydration, followed by hybridization and stringent washes to remove unbound probes. 4′,6-diamidino-2-phenylindole counterstaining was performed to visualize the total DNA. Then, FISH signals were quantified using Adobe Photoshop 2023 to ascertain the chromosomal ploidy and provide a reliable cytogenetic assessment.

### 4.7. Flow Cytometry

Approximately 50 mg of leaves collected from *N. porphyrocoma* (Hance) Bor GXN1 was minced and fixed. Samples were then stained with a DNA-binding fluorescent dye and run on a BD AccuriTMC6 PLUS, which measures the fluorescence (FL) intensity of the stained nuclei. Data were acquired using specialized software that collects histograms of forward scatter, side scatter, and FL signals. The primary threshold was set to exclude debris and noise, whereas secondary thresholds were applied to specific FL channels to ensure accurate data collection.

The resulting histograms were analyzed to determine the relative FL values of the peak positions and coefficient of variation in the G0/G1 peak, providing insights into the integrity of the nuclei and consistency of DNA staining. These values were critical for assessing the samples’ ploidy levels, which were ultimately determined by comparing the FL values of the samples with those of the external standards.

## 5. Conclusions

In this study, we report the first genome assembly of *N. porphyrocoma* (Hance) Bor GXN1, which has a genomic composition of 15 chromosomes. Polyploidization in *N. porphyrocoma* (Hance) Bor occurred at approximately 0.15 Mya. The core ABA signaling pathway gene families using the well-assembled *N. porphyrocoma* (Hance) Bor GXN1 genome are identified and marked on each chromosome. Given this information, *N. porphyrocoma* provides a valuable resource for future breeding programs of sugarcane.

## Figures and Tables

**Figure 1 ijms-26-06124-f001:**
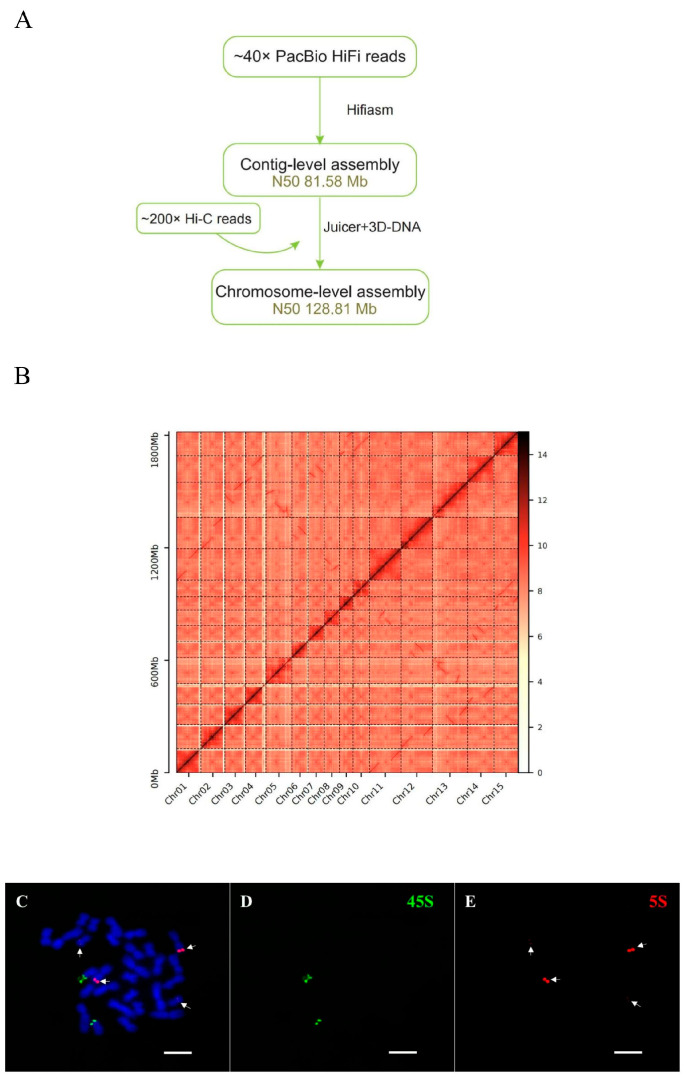
Comprehensive overview of the *N. porphyrocoma* (Hance) Bor GXN1 genome. (**A**) Genomic assembly workflow. (**B**) Hi-C interaction map. (**C**) Chromosomes (blue) detected using 45S rDNA (green) and 5S rDNA (red). Scale: 5 µm. (**D**) Chromosomes detected using 45S rDNA (green). Scale: 5 µm. (**E**) Chromosomes detected using 5S rDNA (red). Scale: 5 µm.

**Figure 2 ijms-26-06124-f002:**
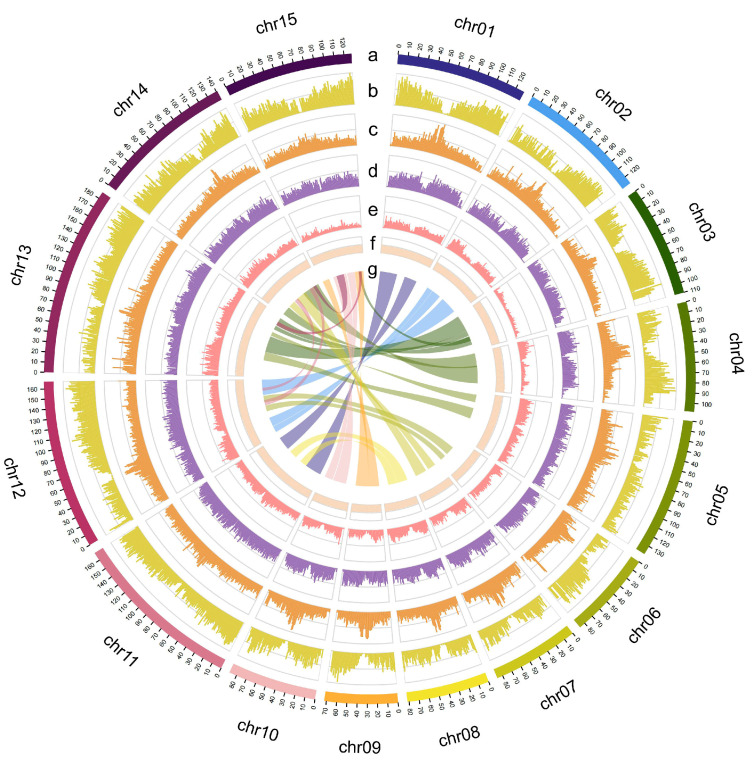
Circos plot of *N. porphyrocoma* (Hance) Bor, from outer to inner sections, the following are separately represented: (a) chromosomes; (b) gene density; (c) Gypsy LTR transposons density; (d) Copia LTR transposons density; (e) DNA transposons density; (f) GC content; (g) collinearity. (b–g) were drawn in 1 Mb sliding windows.

**Figure 3 ijms-26-06124-f003:**
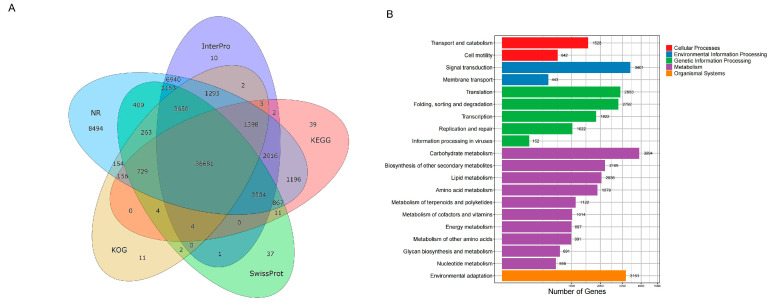
Functional annotation of the protein-coding genes. (**A**) Venn diagram of functional annotation of the protein-coding genes; (**B**) KEGG enrichment analysis.

**Figure 4 ijms-26-06124-f004:**
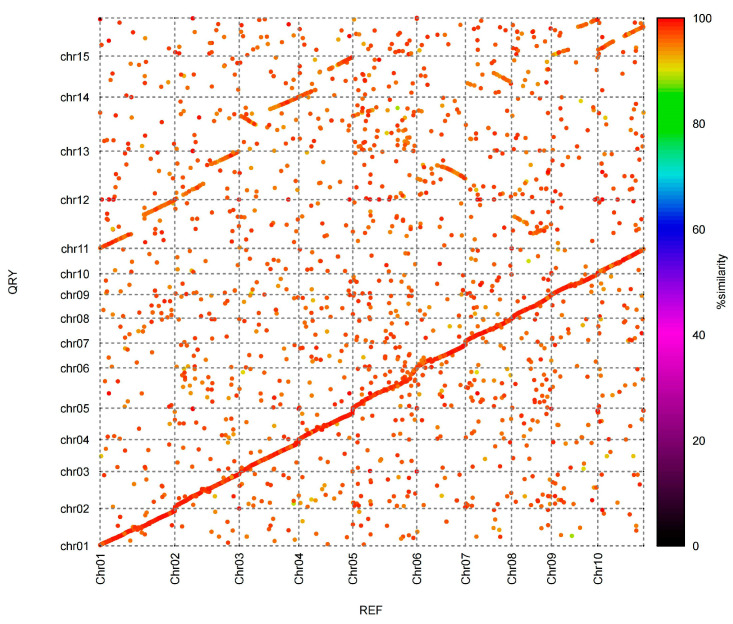
Synteny between *N. porphyrocoma* (Hance) Bor GXN1 and *E. rufipilus*.

**Figure 5 ijms-26-06124-f005:**
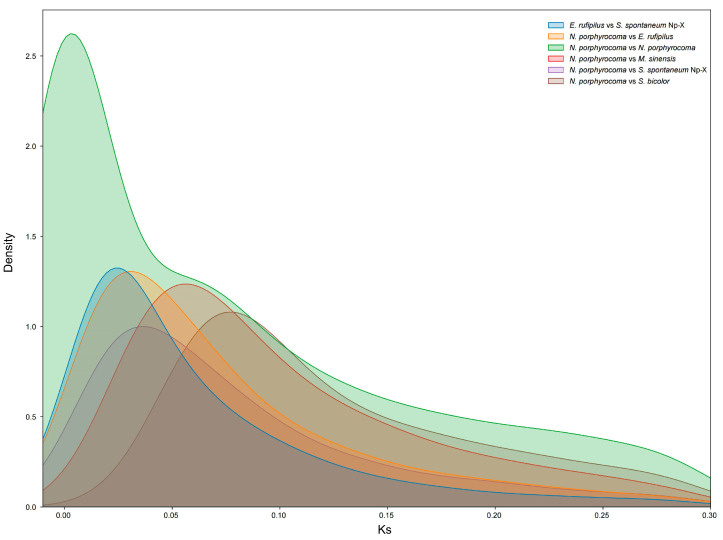
Evolutionary history of *N. porphyrocoma* (Hance) Bor GXN1. The distribution of synonymous nucleotide substitutions (Ks) illustrates the evolutionary divergence within and between species. Colored lines represent the Ks distributions of synonymous gene pairs, indicating genetic divergence between two species or subgenomes within the same species.

**Figure 6 ijms-26-06124-f006:**
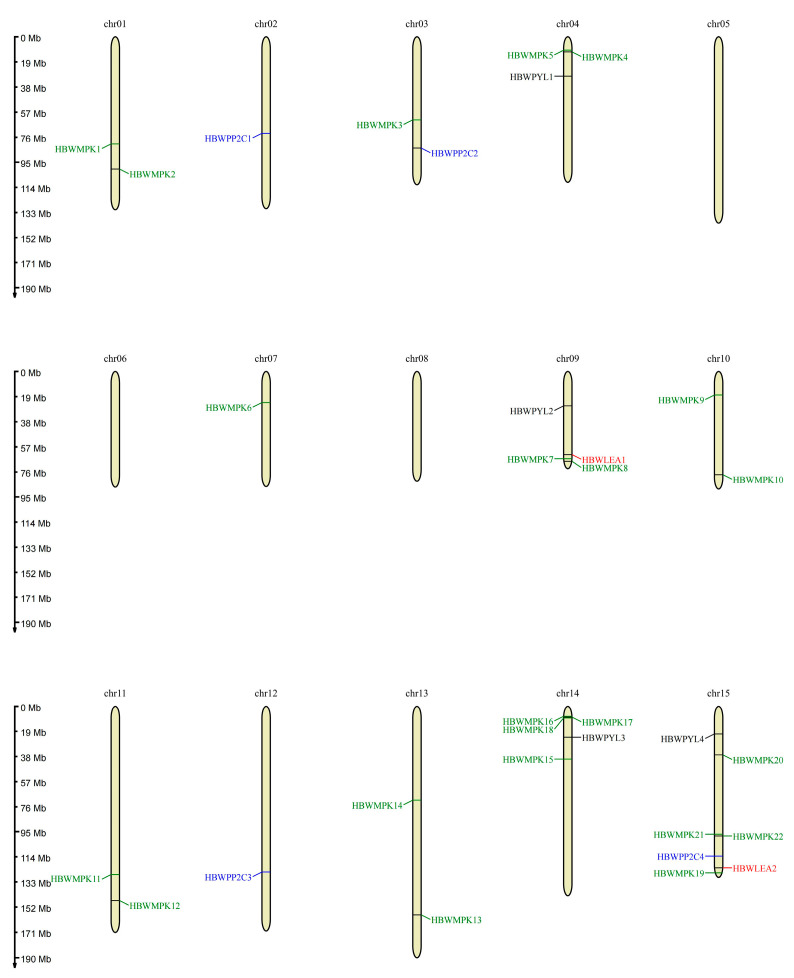
Chromosomal location of ABA signaling pathway genes.

**Table 1 ijms-26-06124-t001:** Genome assembly statistics of *N. porphyrocoma* (Hance) Bor GXN1 genome.

Indicator	Scaffold		Contig	
	Length (bp)	Number	Length (bp)	Number
N50	128,807,855	6	81,578,857	9
N60	127,489,866	8	75,423,851	11
N70	110,254,084	9	66,681,031	14
N80	87,516,119	11	44,940,619	17
N90	85,681,588	13	24,489,615	22
Max length	186,974,133		161,738,000	
Total length	1,822,969,591		1,822,906,591	
Total numbers	-	99	-	225
GC rate	0.46	-	0.46	-
Gap length	63,000	-		-
Anchored to chromosomes	1,818,483,892	15	1,818,420,892	141
BUSCO complete	99.0%			

**Table 2 ijms-26-06124-t002:** Transposable elements classification statistics in the *N. porphyrocoma* (Hance) Bor GXN1 genome.

Type		Length (bp)	% of Genome
Retrotransposons	LTR/Copia	240,473,209	13.19
	LTR/Gypsy	563,894,532	30.93
	LTR/Other	11,340,167	0.62
	SINE	354,429	0.02
	LINE	95,917,621	5.26
	Other	0	0.00
DNA transposons	EnSpm	68,083,692	3.73
	Harbinger	20,544,229	1.13
	hAT	19,054,911	1.05
	Helitron	18,354,173	1.01
	Mariner	1,110	0.00
	MuDR	24,929,917	1.37
	P	156,362	0.01
	Other	17,404,240	0.95
Other	-	2,278,224	0.12
Unknown	-	413,260,571	22.67
Total	-	1,259,966,694	69.12

**Table 3 ijms-26-06124-t003:** Gene prediction statistics of *N. porphyrocoma* (Hance) Bor GXN1.

	Gene Set	Number	Average Gene Length (bp)	Average CDS Length (bp)	Average Exon per Gene	Average Exon Length (bp)	Average Intron Length (bp)
De novo	Augustus	81,024	2603.39	997.29	4.06	245.8	525.34
Homolog	*Zea mays* B37	58,728	3926.02	1332.11	5.19	256.77	619.38
	*Brachypodium stacei*	60,574	3981.24	1276.12	5.04	252.95	668.78
	*Erianthus rufipilus*	81,108	4539.46	1361.4	4.81	283.29	835.08
	*Oryza sativa*	65,992	4162.14	1273.88	4.84	262.96	751.29
	*Saccharum spontaneum* Np-X	107,325	4896.47	1226.1	4.51	271.95	1046.11
	*Setaria italica*	68,011	3906.05	1252.65	4.86	257.57	686.81
	*Sorghum bicolor*	74,180	3692.05	1195	4.67	255.96	680.62
RNA-seq	RNAseq	42,824	6212.9	1177.96	5.61	209.84	1091.34
Final	EVidenceModeler	70,680	3362.29	1136.92	4.67	243.27	605.79

**Table 4 ijms-26-06124-t004:** Divergence time among inter- and intra-species of *N. porphyrocoma*.

Species_Species	Ks	Time (Mya)
*N. porphyrocoma* vs. *N. porphyrocoma*	0.002	0.15
*E. rufipilus* vs. *S. spontaneum* Np-X	0.020	1.54
*N. porphyrocoma* vs. *E. rufipilus*	0.021	1.62
*N. porphyrocoma* vs. *S. spontaneum* Np-X	0.025	1.92
*N. porphyrocoma* vs. *M. sinensis*	0.052	4.00
*N. porphyrocoma* vs. *Sorghum. bicolor*	0.078	6.00

**Table 5 ijms-26-06124-t005:** ABA signaling pathway-related gene families identified in *N. porphyrocoma* (Hance) Bor GXN1 compared with related species.

Species	Ploidy	Genes from Each Genome			
		LEA	MPK	PP2C	PYL
R570	Haplotype	1	8	5	1
CC-01-1940	1x = 10	1	15	7	4
*N. porphyrocoma* (Hance) Bor	2*n* = 2x = 30	2	22	4	4
Erufipilus	2*n* = 2x = 20	1	10	4	2
AP85-441	2*n* = 4x = 32	3	43	13	12
Np-X	2*n* = 4x = 40	2	42	11	8
LA-Purple	2*n* = 8x = 80	3	69	27	14

## Data Availability

The original contributions presented in this study are included in the article/Appendix A. Further inquiries can be directed to the corresponding authors.

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
