# Peer review of "Unveiling the Genome of the Diploid Wild Sugarcane Relative Narenga porphyrocoma (Hance) Bor"

_ijms, 2025, doi:10.3390/ijms26136124_

Round 1
Reviewer 1 Report
Comments and Suggestions for Authors
Line 38 – if using a website as a reference, please follow proper citation format.
Line 43 – the in-text citation format has to be revised to be consistent across the whole manuscript. This is applicable across the whole manuscript.
Line 74 – explain why having fewer chromosomes is considered an advantage when considering crossing
Line 126 - this sentence seems incomplete.
Line 133 – “was” to “were”
Figures 1B – needs a higher resolution image
Figure 2 – an enlarged would be welcomed.
All figures with text/annotation should be reformatted to either increase their size or to be able to show the text clearly.
The results section provided a rich amount of data and it would make this manuscript more impactful if the relevance and implications are further discussed when presenting these results. One suggestion I have is to reorganize the manuscript such that the discussion is incorporated into the results section and the discussion pertaining to individual result can follow the result immediately to give readers a better understanding of the relevance of the results. In turn, this solves another issue with how the discussion section was written – the content there focused more on the general statements of findings but not necessarily linking to the specific data/analysis.
The overall goal of the study strikes the readers as less novel. There is value in providing a holistic genome study of a certain plant but it will be more impactful and engaging if there are more specific reasons for doing so. For example, if the authors can expand the discussion on the advantages of crossing with sugarcane in association with the genome data, the manuscript would become more attractive to researchers outside of the field of genomics or genetics.
Reviewer 2 Report
Comments and Suggestions for Authors
- Materials and Methods, lines 319-323. Please explain in detail the number of plants used to extract DNA. Was the DNA of all of them mixed in a bulk?
- A somewhat detailed description of the methodology applied to obtain results is provided in different parts of the Results section. In order to make the reading more fluid and not lose consistency, it is suggested to place as much as possible of the methodology used in the Materials and Methods section. Only take care that context is not lost when doing so.
- Conclusion, lines 427-428. The authors state that two-thirds of drought resistance genes are regulated by ABA Is this result directly derived from the present study or was it taken from literature? If it was taken from literature, it should not remain in the Conclusions section.
